# Mechanism of Cadmium Exposure Induced Hepatotoxicity in the Mud Crab (*Scylla paramamosain*): Activation of Oxidative Stress and Nrf2 Signaling Pathway

**DOI:** 10.3390/antiox11050978

**Published:** 2022-05-17

**Authors:** Changhong Cheng, Hongling Ma, Guangxin Liu, Sigang Fan, Zhixun Guo

**Affiliations:** Key Laboratory of South China Sea Fishery Resources Exploitation & Utilization, Ministry of Agriculture and Rural Affairs, South China Sea Fisheries Research Institute, Chinese Academy of Fishery Sciences, Guangzhou 510300, China; chengchanghong09@163.com (C.C.); maling.5679@163.com (H.M.); Guangxin_liu1988@163.com (G.L.); fansigang@scsfri.ac.cn (S.F.)

**Keywords:** *Scylla paramamosain*, Nrf2 signaling pathway, oxidative stress, hepatotoxicity

## Abstract

Cadmium, one of the most toxic heavy metals, can cause severe oxidative damage to aquatic animals. However, the mechanism whereby the mud crabs respond to cadmium exposure remains unclear. This study investigated the effects of cadmium exposure on oxidative stress and histopathology changes and evaluated the role of the Nrf2 signaling pathway in regulating responses to cadmium-induced hepatotoxicity were investigated in mud crabs. Mud crabs were exposed to 0, 0.01, 0.05, and 0.125 mg/L cadmium for 21 d. The present results indicated that cadmium exposure increased hydrogen peroxide (H_2_O_2_) production, lipid peroxidation and tissue damage, but decreased the activity of superoxide dismutase (SOD) and catalase (CAT), and caused lipid peroxidation and tissue damage. The results of an integrated biomarker index analysis suggested that the toxicity of cadmium was positively related to cadmium concentration. The expression levels of the Nrf2 signaling pathway (Nrf2, metallothionein, and cytochrome P450 enzymes) were up-regulated after cadmium exposure. Silencing of Nrf2 in vivo decreased antioxidant gene (SOD, CAT, and glutathione S-transferase) expression, suggesting that Nrf2 can regulate antioxidant genes. Knocking down Nrf2 in vivo also significantly decreased the activity of SOD and CAT after cadmium exposure. Moreover, silencing of Nrf2 in vivo enhanced H_2_O_2_ production and the mortality rates of mud crabs after cadmium exposure. The present study indicated that cadmium exposure induced hepatotoxicity in the mud crab by increasing H_2_O_2_ content, which decreased the antioxidant capacity, leading to cell injury. In addition, the Nrf2 is activated to bound with antioxidant response element, initiating the expression of antioxidant enzyme genes during cadmium induced hepatotoxicity in the mud crabs.

## 1. Introduction

Heavy metal pollution in aquatic environments has serious harmful effects on living organisms. Heavy metals from mine wastes, fertilizer use, and industrial waste are the main sources of pollution in the aquatic environment. Heavy metals can be easily accumulated in organisms, which might ultimately affect human health [1]. Among these heavy metal pollutants, cadmium is a nonessential heavy metal that has high toxic effects on the health of humans and the environment. In natural surface water, the level of cadmium concentration is very low. However, a high level of cadmium concentration in many areas of China can reach as high as 100–900 μg/L [2]. The toxicity of cadmium can harm development and reproduction, decrease growth and survival, and damage tissues and organs [3,4,5]. Although many reports have shown the toxic effects of cadmium on aquatic animals, the mechanisms involved are not well understood.

It has been reported that reactive oxygen species (ROS) can be induced by cadmium exposure, which might cause oxidative stress, leading to damage to macromolecules, such as DNA and proteins [6,7]. To prevent the formation of ROS, organisms develop an antioxidant system, which consists of antioxidant enzymes. Nrf2 is a basic leucine zipper nuclear transcription factor that plays a key role in response to various environmental pollutants [8]. It is responsible for regulating the expression of a variety of antioxidant genes [9]. Therefore, activation of the Nrf2 signaling pathway can regulate antioxidant and detoxification cellular responses [10]. Nrf2 might participate in host defense during antibacterial immunity response [11]. Many studies have also shown that the activation of the Nrf2 pathway could protect against hepatotoxicity induced by heavy metal pollution [12,13]. However, the role and the regulatory mechanism of the Nrf2 signaling pathway in crustaceans against cadmium exposure still need further study.

Cytochrome P450 enzymes (CYPs) are a hemoglobin superfamily that plays a crucial role in the degradation and detoxification of xenobiotics. Activation of CYPs is involved in responses to oxidative stress [14]. A previous study showed that cadmium exposure could induce the expression of CYPs [15]. In addition, CYPs have been used as a sensitive biomarker for heavy metal exposure [16]. Metallothionein (MT) is a low molecular weight protein that contains cysteine. MT is also involved in transporting various metals [17]. It plays an important role in the detoxification of metals and the scavenging of free radicals [18]. The induction of MT expression has been widely applied as a biomarker for the toxicity of heavy metals [19].

Mud crab (*Scylla paramamosain*) is an important commercial crustacean that is widely distributed in the warm temperate zones of Southeast Asia. It has become more and more popular in Chinese markets due to its high quality and fast growth. It is also exposed to various pollutants due to its habitat and diet. Therefore, understanding the mechanisms of detoxification might lead to the development of ways to protect mud crabs against environmental stressors. In this study, the effects of cadmium exposure on the antioxidant system and resulting histopathological alterations were investigated. To understand the Nrf2 signaling pathway in cadmium-induced hepatotoxicity in mud crabs, mRNA levels of Nrf2, CYP2, and MT after cadmium exposure were examined. Furthermore, we used RNAi technology to determine the function of the Nrf2 signaling pathway in response to cadmium exposure. This study will help us to understand the defense mechanism of crustaceans against environmental stress.

## 2. Materials and Methods

### 2.1. Mud Crabs

Mud crabs (51 ± 3 g) were collected from a mud crab farm (Taishan, China) and acclimated in tanks with circulating seawater (5 ppt salinity) at 25 ± 1 °C. During the acclimation period, they were fed a commercial diet twice a day.

### 2.2. Cadmium Exposure

Cadmium chloride was obtained from Sigma (Sigma, St. Louis, MO, USA). In our preliminary experiments, we found that 96 h LC50 of cadmium on mud crabs was approximately 10 mg/L. A stock solution of cadmium chloride (1 g/L) was used as a source of cadmium and was subsequently diluted to the desired concentrations. In the present study, groups of mud crabs were treated with 0 (control), 0.01 (1/1000 96 h-LC50), 0.05 (1/200 96 h-LC50), or 0.125 (1/200 96 h-LC50) mg/L Cd^2+^. Each group included three tanks, and each tank contained 20 mud crabs. The water was changed daily. Water samples were acidified with HNO_3_. Cadmium concentrations in water were quantified with ICP-OES (Optima 8300, Perkin Elmer Instruments, Waltham, MA, USA) [20]. Water quality parameters were as follows: salinity 5 ppt, temperature 25 ± 1 °C, and total ammonia < 0.05 mg/L. After 21 days of challenge, nine mud crabs from the control group and each cadmium treatment group were sampled. Hepatopancreas tissues were collected for the oxidative stress study.

### 2.3. Measurement of Oxidative Stress Parameters 

Six hepatopancreas tissues from each group were homogenized in a homogenization buffer. After tissue homogenization, cell suspensions were centrifuged at 3000× *g* for 15 min. Finally, the supernatants were collected for the analyses of biochemical parameters. The measurement of superoxide dismutase (SOD), catalase (CAT), total antioxidant capacity (T-AOC), hydrogen peroxide (H_2_O_2_), and malondialdehyde (MDA) was performed using the corresponding assay kits (Nanjing Jiancheng Bioengineering Institute, Nanjing, China) according to the manufacturer’s protocols. SOD and CAT were analyzed according to the method of Jia et al. [21]. H_2_O_2_ content was measured according to the method of Lin et al. [22]. MDA content was assayed according to the method of Ohkawa et al. [23] using the thiobarbituric acid reactive species assay at 532 nm. T-AOC was determined by the reduction of a ferric tripyridyltriazine complex to ferrous tripyridyltriazine [24].

### 2.4. Comprehensive Toxicity Assessment

An integrated biological responses version 2 (IBRv2) analysis was used to assess the toxicity of cadmium as described by Sanchez et al. [25]. An IBRv2 value was calculated based on biomarkers, including SOD, CAT, T-AOC, H_2_O_2_, and MDA. An IBRv2 value can provide a better understanding of the harmful effects of cadmium exposure.

### 2.5. Histopathologic Investigation

Six hepatopancreas tissues from each group were immediately fixed in 10% formaldehyde, embedded in paraffin, and stained with hematoxylin and eosin. Then, the stained sections were viewed under a light microscope (Olympus, Tokyo, Japan).

### 2.6. Quantitative Real-Time PCR (qRT-PCR) Analysis

Total RNA was extracted from the hepatopancreas using RNAiso Plus reagent (Takara, Dalian, China) according to the manufacturer’s recommendations. The quality of the RNA was verified by agarose electrophoresis. Single-strand cDNA was synthesized using a PrimeScriptTM reverse transcriptase kit (Takara, Dalian, China). A qRT-PCR was performed to measure the expression of Nrf2, CYP2, and MT (Appendix A). The qRT-PCR was carried out in a Qtower96G real-time system (Jena, Germany) using SYBR Green. The reaction mixtures were 20 μL, containing 2 μL of diluted cDNA sample (50 ng/μL), 10 μL 2 × SYBR Premix Ex Taq, and 0.4 μL each of primer (10 μM) and 7.6 μL dH_2_O. The qRT-PCR conditions were as follows: 94 °C for 10 min, then 45 cycles at 95 °C for 30 s or 60 °C for 30 s. 18S rRNA was utilized as a reference gene. All qRT-PCRs were performed at least in triplicate. The relative expression of each gene was determined via the 2^−^^ΔΔCT^ method [26].

### 2.7. Nrf2 Silencing in Mud Crabs

For gene knockdown experiments, double-stranded RNA Nrf2 and green fluorescent protein (GFP) were amplified by PCR using gene-specific primers (Appendix A). DsNrf2 and dsGFP were synthesized using the Transcription T7 Kit (Promega, Madison, WI, USA) following the manufacturer’s protocol. Sixty mud crabs were separated into two groups. One group was injected with 50 ug dsNrf2, and the other group was injected with an equivalent of dsGFP. At 48 h after injection, six hepatopancreas tissues from each group were sampled for RNA extraction. To explore the functions of Nrf2, the mRNA expressions of antioxidant enzyme genes (SOD, CAT, GST, and Gpx3) were investigated in both the Nrf2-silenced group and the control (dsGFP) group. A qRT-PCR was performed as described above.

### 2.8. Cadmium Exposure in Nrf2-Silenced Mud Crabs

To understand the effects of cadmium exposure on mud crabs after Nrf2 gene silencing, 120 mud crabs were injected with dsNrf2 and dsGFP, respectively. At 48 h after injection, both groups were acutely exposed to 2.5 mg/L cadmium (1/4 96 h-LC50). Six hepatopancreas tissues from each group were collected at 0, 6, 12, and 24 h after cadmium exposure. Then, these hepatopancreas tissues were used to measure SOD, CAT, T-AOC, and H_2_O_2_. Measurements of these oxidative stress parameters were detected as described in Section 2.3. Meanwhile, the numbers of dead mud crabs in both the Nrf2-silenced group and the control group were recorded after cadmium exposure.

### 2.9. Statistical Analysis

The experimental data were presented as means ± standard deviation. For all analyses, Levene and Shapiro–Wilk tests were used to verify the homogeneity and normality of variances, respectively. Differences between groups were analyzed using a one-way ANOVA followed by Duncan’s multiple range tests in SPSS 18.0 software (SPSS; Chicago, IL, USA). A *p*-value < 0.05 was considered a significant difference.

## 3. Results

### 3.1. Oxidative Stress and IBRv2 after Cadmium Exposure

As shown in Figure 1A, SOD activity did not change in the treatment with 0.01 mg/L cadmium and significantly decreased in the treatments with 0.05 and 0.125 mg/L cadmium. CAT activity significantly decreased in the treatments with 0.01, 0.05, and 0.125 mg/L cadmium when compared with that in the control group (Figure 1B). T-AOC level was not significantly affected by cadmium exposure (Figure 1C). H_2_O_2_ content in the treatments with 0.05 and 0.125 mg/L cadmium was significantly higher than that in the control group (Figure 1D). Cadmium exposure strongly enhanced the levels of MDA in the 0.01, 0.05, and 0.125 mg/L cadmium treatments when compared with that in the control group. The highest levels of MDA were observed in the 0.05 and 0.125 mg/L cadmium treatments.

Based on the above-described oxidative stress biomarkers, IBRv2 values were calculated (Appendix A). IBRv2 values were positively correlated with cadmium concentrations. The highest IBRv2 value was 7.76 in the 0.125 mg/L cadmium group.

### 3.2. Histopathological Changes after Cadmium Exposure

Histopathological changes after cadmium exposure are shown in Figure 2. The hepatopancreas tissues of the control group displayed a normal histological appearance (Figure 2A). However, these hepatopancreas tissues showed some signs of cell border diffusion and cytoplasmic vacuolization when exposed to 0.01 mg/L cadmium (Figure 2B). In addition, some signs of cell abscission and cell lysis were observed in the 0.05 and 0.125 mg/L cadmium treatments (Figure 2C,D).

### 3.3. Effects of Cadmium Exposure on the Expression of Nrf2, CYP2, and MT

As shown in Figure 3A, the expression level of Nrf2 significantly increased after cadmium exposure. The highest expression level of Nrf2 was observed in the 0.05 mg/L cadmium treatment. The transcription level of CYP2 significantly increased in the treatments with 0.01, 0.05 and 0.125 mg/L cadmium (Figure 3B). The expression level of MT in the treatments with 0.01, 0.05, and 0.125 mg/L cadmium was significantly higher than that in the control group (Figure 3C).

### 3.4. The Expression of Genes Related to Antioxidants after Knockdown of Nrf2

To understand the possible mechanism of Nrf2 in regulating antioxidant-related genes in mud crabs, Nrf2 was knocked down by the injection of Nrf2 dsRNA. The expression levels of Nrf2, SOD, GST, CAT, and Gpx3 are shown in Figure 4. The transcription level of Nrf2 in the dsNrf2 group at 48 h significantly decreased by 84% when compared with that in the control group. The expression levels of SOD, CAT, and GST in the dsNrf2 group were significantly lower than those in the dsGFP group after dsRNA injection. Compared to the control group, the change in Gpx3 showed no significant difference in the dsNrf2 group after dsRNA injection.

### 3.5. Effects of Nrf2-Interfered on Oxidative Stress Biomarkers after Cadmium Exposure

As shown in Figure 5A, SOD activity in the control group significantly decreased at 6 and 12 h after cadmium exposure. SOD activity in the dsNrf2 group was significantly lower than in the dsGFP group at 0, 6, and 24 h after cadmium exposure. CAT activity in the control group significantly decreased at 12 h after cadmium exposure. CAT activity was lower at 0 and 6 h after cadmium exposure in the dsNrf2 group than in the control group (Figure 5B). T-AOC levels in both the dsNrf2 group and the dsGFP group did not change after cadmium exposure (Figure 5C). H_2_O_2_ content in the control group significantly increased from 6 to 24 h after cadmium exposure. H_2_O_2_ content was higher at 0 and 24 h after cadmium exposure in the dsNrf2 group than in the control group (Figure 5D).

### 3.6. The Mortality of Mud Crabs after Cadmium Exposure

The survival rates of mud crabs in the dsNrf2 group and dsGFP group after cadmium exposure are shown in Figure 6. Cumulative mortality of mud crabs was 87% in the dsNrf2 group and 63% in the control group at 60 h after cadmium exposure. Therefore, knocking down Nrf2 in vivo significantly decreased the survival rate of mud crabs after cadmium exposure.

We also demonstrated a plausible mechanism of the Nrf2 signaling pathway in cadmium-induced hepatotoxicity in mud crabs (Figure 7).

## 4. Discussion

Cadmium, one of the most toxic heavy metals, has negative effects on aquatic animals. However, the mechanisms of detoxification and antioxidant defense against cadmium exposure in crustaceans are still unclear. The Nrf2 signaling pathway is a master regulator of cellular responses by playing a crucial role in cellular detoxification against environmental stresses. In this study, we investigated the mechanism of the Nrf2 signaling pathway in response to cadmium stress in mud crabs.

Acute or chronic exposure to cadmium is mainly evident in the hepatopancreas. Excessive cadmium can damage hepatopancreas function. Cadmium is considered the inducer of ROS, causing oxidative stress [27]. Our study suggested that chronic exposure to 0.05 and 0.125 mg/L of cadmium induced H_2_O_2_ (a major by-product of ROS) production. The overproduction of ROS can promote protein and DNA damage. The antioxidant system is considered the first line of defense to remove surplus ROS. SOD and CAT are the most important antioxidant enzymes for scavenging superoxide and hydroxyl radicals. A previous study showed that cadmium suppressed the antioxidant enzyme activities and damaged the antioxidant defense system [7]. We found that chronic cadmium exposure decreased the activities of SOD and CAT and increased H_2_O_2_ content. This result indicated that a reduction in antioxidant capacity could cause lipid peroxidation. MDA is the end-product of lipid peroxidation. Elevated MDA levels have been reported in *Eriocheir sinensis* after cadmium exposure [22]. Similar results were also observed in this study, suggesting that chronic exposure to cadmium can cause severe oxidative damage. The tissue damage induced by cadmium was also marked by histopathological changes. Zhang et al. [20] reported that cadmium exposure caused obvious histological changes, including tubule lumen dilatation and epithelium vacuolization, in *Procambarus clarkii*. Our study found that histological changes after cadmium exposure included signs of cell lysis and abscission, suggesting that cadmium could damage the normal cellular structure and affect normal physiological function. Moreover, the IBRv2 analysis also provided supporting evidence that the toxicity of cadmium is positively related to cadmium concentration.

The toxicity of cadmium is involved in the induction of host defenses. CYPs are monooxygenases that play a protective role in organisms against oxidative damage. A previous study reported that the mRNA levels of CYPs in common carp were increased after cadmium exposure, suggesting the activation of detoxification [15]. Our results showed that CYP2 was induced after cadmium exposure, indicating that detoxification is involved in cadmium-induced toxicity in mud crabs. MT, a group of ubiquitous low-molecular-weight proteins, is a promising candidate for heavy metal detoxification. MT is involved in the transport and storage of metals [17]. It can scavenge ROS through the oxidation of cysteine thiol groups. Activation of MT is involved in detoxification after heavy metal exposure [28]. The present study showed that the expression level of MT was strongly induced by cadmium exposure, which can alleviate the toxicity of cadmium.

Nrf2, a central regulator of cellular responses, plays a crucial role in coping with oxidative stress. The lack of Nrf2 can cause hepatic lipid peroxidation and hepatic injury [29]. Under normal physiological conditions, Nrf2 expression is controlled by Keap1. Under oxidative stress, Nrf2 is uncoupled from Keap1 and accumulated in the nucleus, which is bound with the antioxidant response element, initiating the expression of antioxidant enzyme genes [30]. Activated Nrf2 can mediate some signaling proteins to regulate innate immune responses, apoptosis, antioxidant function, and drug metabolism [9]. Dong et al. [31] reported that cadmium exposure activated Nrf2, which promoted antioxidant gene transcription. A previous study showed that Nrf2 activation prevented cadmium-induced acute liver injury [32]. The present study found that the transcription level of Nrf2 was up-regulated after cadmium exposure, suggesting that Nrf2 was activated to protect mud crabs against hepatopancreas damage induced by cadmium.

To further explore the regulation of the Nrf2 signaling pathway in mud crabs, we knocked down Nrf2 by injecting dsRNA. The expression levels of Nrf2 and some antioxidant enzyme genes, including SOD, CAT, and GST, significantly decreased in the Nrf2-interfered group when compared with the control group. Wang et al. [33] reported that after knocking down Nrf2 expression in clams, antioxidant gene expression was changed. Furthermore, we found that knocking down Nrf2 in mud crabs significantly reduced the activities of SOD and CAT and increased H_2_O_2_ content. All of these results suggest that Nrf2 is required for the induction of antioxidant defense genes.

To verify the function of Nrf2 involving in antioxidant defense against cadmium stress, we investigated the effects of Nrf2 silencing on the antioxidant enzymatic activities, H_2_O_2_ contents and cumulative mortality rates of mud crabs after cadmium exposure. The present study showed that H_2_O_2_ content in the Nrf2-interfered group was much higher than that in the control group at 24 h after cadmium exposure. These results might be evidence that knocking down Nrf2 inhibited gene expression as an antioxidant defense. Accumulation of ROS can damage cells, leading to cell death. The present study showed that the mortality of mud crabs in both the Nrf2-interfered and the control group significantly increased after cadmium exposure. This result is consistent with that obtained in a previous study [34]. The high mortality might be associated with higher H_2_O_2_ content and serious cell death after cadmium exposure. Moreover, after knocking down Nrf2, the mortality of mud crabs significantly increased after cadmium exposure. These results are attributed to that Nrf2 silencing may impair the antioxidant modulation of the mud crab exposed to cadmium. Therefore, all these results further confirm that the Nrf2 signaling pathway plays a vital role in mechanisms of detoxification and antioxidant defense against oxidative damage induced by cadmium exposure.

## 5. Conclusions

The present study demonstrated a plausible mechanism of the Nrf2 signaling pathway in cadmium-induced hepatotoxicity in mud crabs (Figure 7). The results showed that cadmium exposure increased H_2_O_2_ content and decreased antioxidant capacity, causing severe tissue damage. In addition, the Nrf2 signaling pathway and detoxification were activated in cadmium-induced hepatotoxicity, which suggests they have essential roles in cadmium exposure. RNAi technology was used to investigate the antioxidant defense mechanisms of the Nrf2 signaling pathway. Knocking down Nrf2 in vivo significantly decreased the antioxidant capacity and increased the mortality of mud crabs after cadmium exposure. This study provided a novel insight into mechanisms of detoxification and antioxidant defense against cadmium exposure in crustaceans.

## Figures and Tables

**Figure 1 antioxidants-11-00978-f001:**
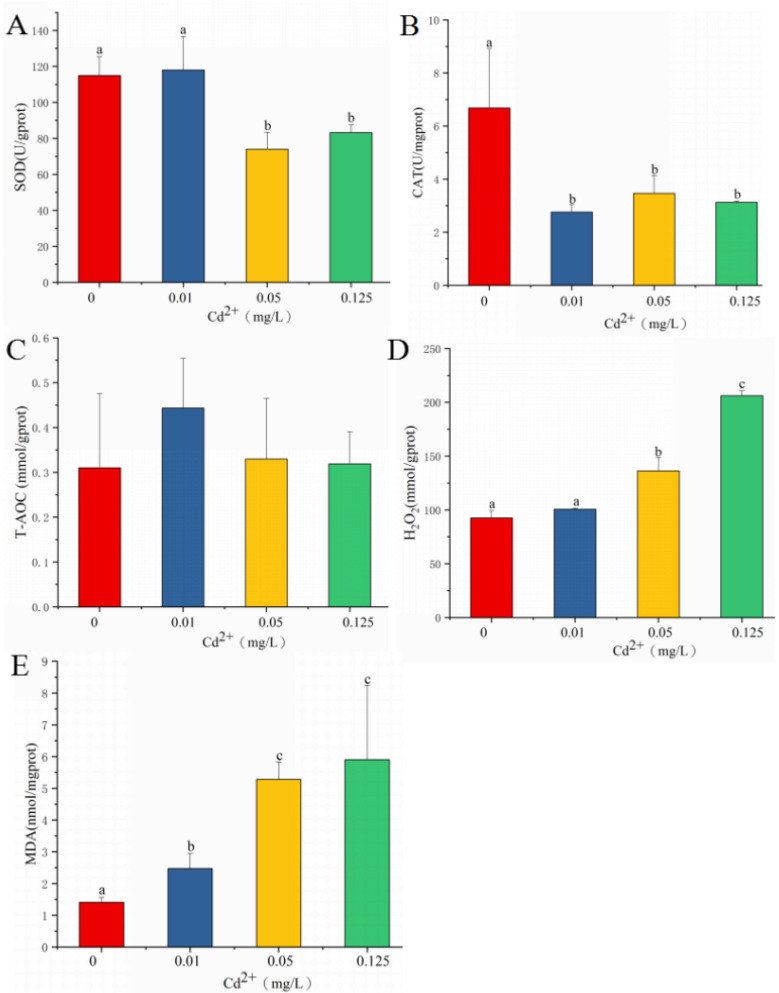
SOD (**A**), CAT (**B**), T-AOC (**C**), H_2_O_2_ (**D**), and MDA (**E**) after cadmium exposure. Values are expressed as the mean ± SD. Different letters reflect significant differences between the control group and the cadmium treatment groups (*p*
*<* 0.05).

**Figure 2 antioxidants-11-00978-f002:**
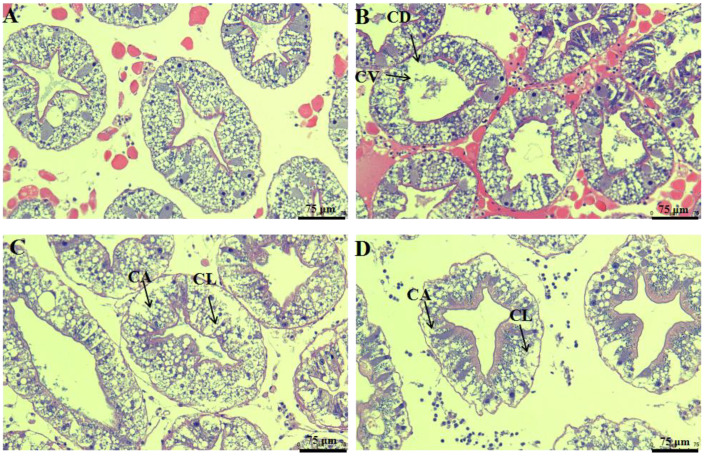
Histological changes after cadmium exposure: (**A**) control group; (**B**) 0.01 mg/L cadmium treatment group; (**C**) 0.05 mg/L cadmium treatment group; (**D**) 0.125 mg/L cadmium treatment group. Cell abscission (CA); Cytoplasmic vacuolization (CV); Cell border diffusion (CD); Cell lysis (CL).

**Figure 3 antioxidants-11-00978-f003:**
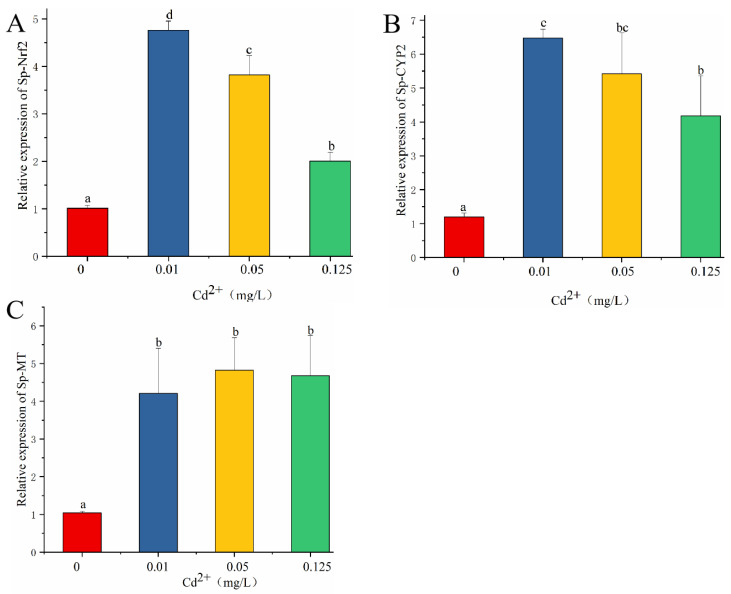
Effects of cadmium exposure on the expression levels of Nrf2 (**A**), CYP2 (**B**), and MT (**C**). Values are expressed as the mean ± SD. Different letters reflect significant differences between the control group and the cadmium treatment groups (*p*
*<* 0.05).

**Figure 4 antioxidants-11-00978-f004:**
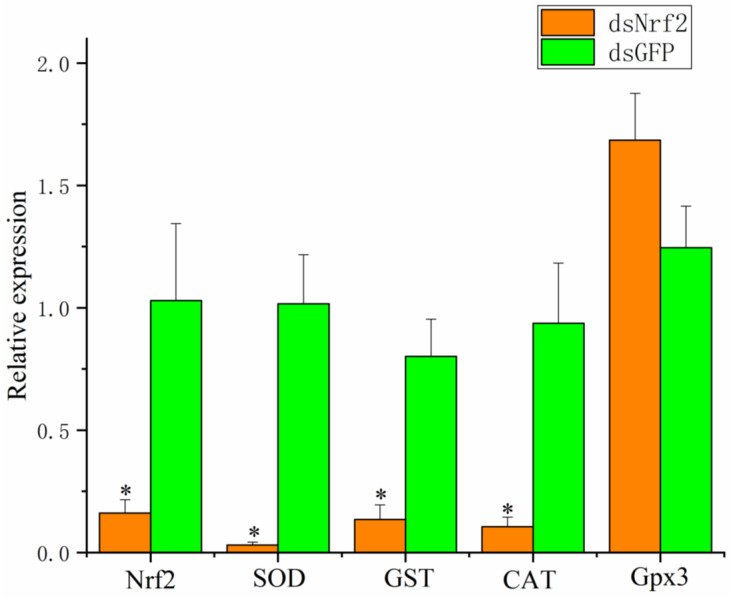
Expression levels of Nrf2, SOD, GST, CAT, and Gpx3 at 48 h after dsRNA injection. Values are expressed as the mean ± SD. Asterisks reflect significant differences between the dsNrf2 group and the dsGFP group (* *p*
*<* 0.05).

**Figure 5 antioxidants-11-00978-f005:**
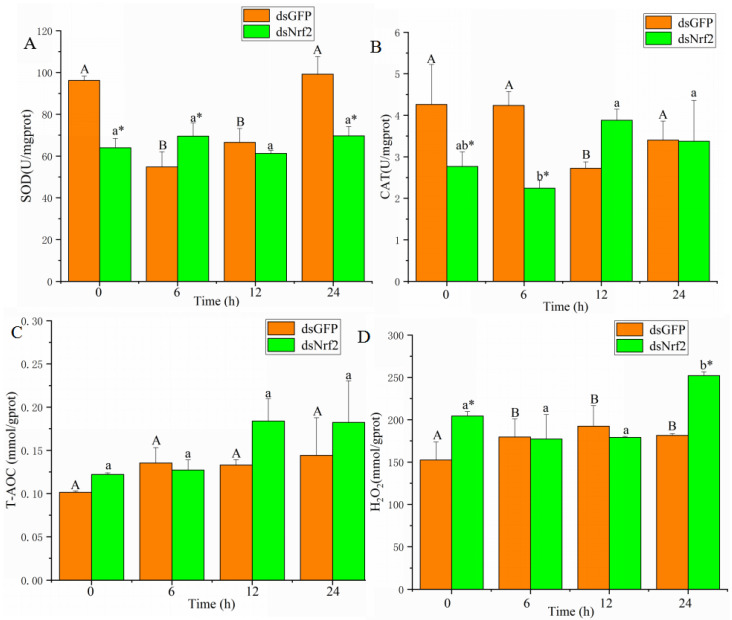
SOD (**A**), CAT (**B**), T-AOC (**C**), and H_2_O_2_ (**D**) in the dsNrf2 group and the dsGFP group after cadmium exposure. Values are expressed as the mean ± SD. Significant differences between the dsNrf2 group and the dsGFP group at each time are indicated by asterisks. Times in each group that have different letters were significantly different (* *p <* 0.05).

**Figure 6 antioxidants-11-00978-f006:**
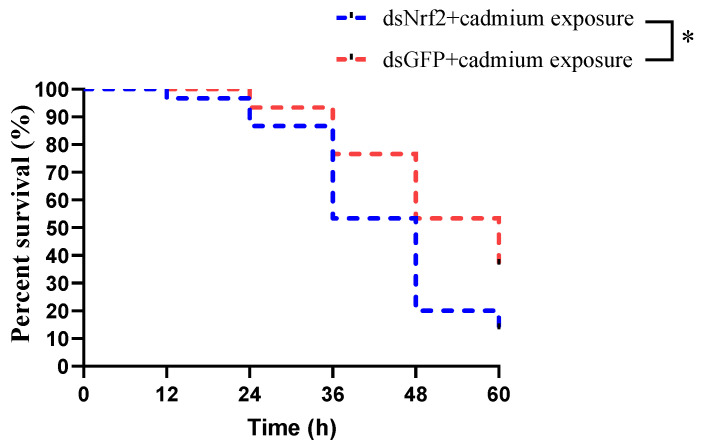
The percentage survival of mud crabs in the dsNrF2 group and dsGFP group (the dsGFP group was used as the control) after cadmium exposure. Asterisks reflected significant differences between the dsNrf2 group and the dsGFP group.

**Figure 7 antioxidants-11-00978-f007:**
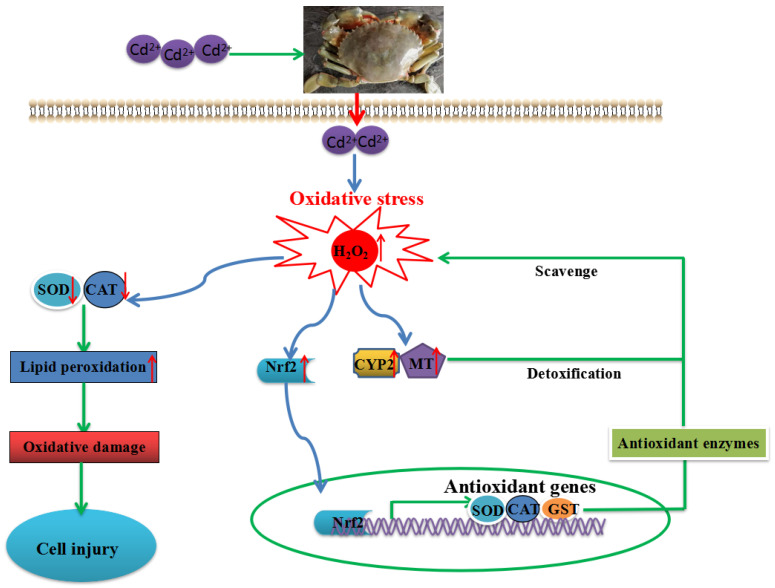
The molecular mechanism of the Nrf2 signaling pathway in cadmium-induced hepatotoxicity in mud crabs.

## Data Availability

Data are contained within the article and Appendix A.

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
