# Peer review of "Mechanism of Cadmium Exposure Induced Hepatotoxicity in the Mud Crab (Scylla paramamosain): Activation of Oxidative Stress and Nrf2 Signaling Pathway"

_antioxidants, 2022, doi:10.3390/antiox11050978_

Round 1
Reviewer 1 Report
antioxidants-1696894
Title
Mechanism of cadmium exposure induced hepatotoxicity in the mud crab (Scylla paramamosain): Activation of oxidative stress and Nrf2 signaling pathway
Authors: Changhong Cheng, Hongling Ma, Guangxin Liu, Sigang Fan, Zhixun Guo
Cheng et al performed a study about the toxic effects of the exposure to cadmium in the hepatopancreas of mud crabs. They observed that cadmium induces oxidative stress and strongly alters protein defence response in liver. Moreover, they could demonstrate that the Nrf2 signalling regulates the expression of scavenger and detoxification systems.
All these data give an important contribute to understand the toxic effects of the cadmium pollution impact in aquatic animals and the interference of Nrf2 signalling pathway in the antioxidant defence. However, the manuscript needs to be revised and some issues should be better argued.
In the introduction the phrase in line 61, page 2, needs to be referenced.
In general, the methods should be detailed in every section, and the number of crabs assayed in each group should be indicated, the statical analysis must be better described: what multiple comparison test did authors apply?
In results, what does the total antioxidant capacity measure? Please explain.
Figure 2. The histopathological changes must be described in detail and indicated in each image of the figure.
Authors observed that cadmium exposure induces the gene expression of some enzymes involved in the detoxification system (section 3.3.): did they evaluate protein expression of these enzymes? How do they explain that the Gpx3 is unaffected?
Figure 5: After 24 hours, in dsNfrf2 group, H2O2 production increases while the activity of scavenger enzymes (SOD and CAT) is unchanged, how do authors interpret this difference?
Why were Nrf2-silenced mud crabs exposed to 2,5mg/L cadmium? Furthermore, carefully revise all section 3.6.
Minor comments
A list of abbreviation should be presented.
The legend of figure 2 should be revised.
Figure 6: please label x-axis in the chart (h).
Author Response
We would like to express our sincere thanks to you for the constructive and positive comments. Point by point responses to the comments are listed below.
1.In the introduction the phrase in line 61, page 2, needs to be referenced.
Response: Thank you for your suggestions. We have added the referenced in the paper (line 62, line 395-396).
2.In general, the methods should be detailed in every section, and the number of crabs assayed in each group should be indicated, the statical analysis must be better described: what multiple comparison test did authors apply?
Response: Thank you for your suggestions. We have revised them in section 2. For all analyses, Levene's and Shapiro-Wilk's tests were used to verify the homogeneity and normality of variance, respectively. Differences among groups were analyzed by a one-way ANOVA followed by Duncan's multiple range tests by SPSS 18.0 software (SPSS; Chicago, IL). P value < 0.05 was considered as the significant difference. We added this information in the paper (line 150-153).
3.In results, what does the total antioxidant capacity measure? Please explain.
Response: Total antioxidant capacity can reflect the total antioxidant level, which is composed of various antioxidant substances and antioxidant enzymes, such as antioxidant enzymes, vitamin C, vitamin E and carotene, etc. Total antioxidant capacity was assayed by using ferric reducing/antioxidant power assay. It depends upon the reduction of a ferric tripyridyltriazine (Fe3+-TPTZ) complex to the ferrous tripyridyltriazine (Fe2+-TPTZ). Fe2+-TPTZ has an intensive blue color and can be monitored at 593 nm. The ability of a compound to produce Fe2+ from Fe3+ defined as “antioxidant power”.
4.Figure 2. The histopathological changes must be described in detail and indicated in each image of the figure.
Response: Thank you for your suggestions. We have revised them (line 175-178).
5.Authors observed that cadmium exposure induces the gene expression of some enzymes involved in the detoxification system (section 3.3.): did they evaluate protein expression of these enzymes?
Response: Thank you for your suggestions. Due to the lack of detoxification enzymes of antibody, we did not measure the expression of Nrf2, CYP2, and MT after cadmium exposure at protein level. Thus, we investigated the mRNA expression levels of Nrf2, CYP2, and MT in responses to cadmium exposure at gene level.
6.How do they explain that the Gpx3 is unaffected?
Response: Thank you for your suggestions. Glutathione peroxidase (GPx) is an important antioxidant enzyme. Several studies have confirmed that Nrf2 could induce expression of an array of antioxidant response element-dependent genes, such as GPx in human. GPx contains GPx1, GPx2, GPx3, GPx4, GPx5, GPx6, GPx7 and GPx8. In this study, knocking down Nrf2 can not affect Gpx3 expression. The results implied that the regulation of Gpx3 expression may not via the Nrf2 signal pathway in the mud crab. The function of Nrf2 regulating GPx expression between human and mud crab may be difference.
7.Figure 5: After 24 hours, in dsNfrf2 group, H2O2production increases while the activity of scavenger enzymes (SOD and CAT) is unchanged, how do authors interpret this difference?
Response: Thank you for your suggestions. In the present study, SOD activity in the dsNrf2 group were significantly lower than that in the dsGFP group at 24 h after cadmium exposure. CAT activity in both dsNrf2 and dsGFP group did not change at 24 h after cadmium exposure. However, compared to the control group, H2O2 content in dsNrf2 group were higher at 24 h after cadmium exposure. The reason may be attributed to that Nrf2 can also control other antioxidant response element-dependent genes, such as Prx and GST. We speculated that knocking down Nrf2 may decrease these antioxidant enzymes levels, causing H2O2 content increased after cadmium exposure.
8.Why were Nrf2-silenced mud crabs exposed to 2.5mg/L cadmium? Furthermore, carefully revise all section 3.6.
Response: In the present study, we want to investigate the regulation of Nrf2 signaling pathway in the mud crab after acute cadmium exposure. Furthermore, it is helpful to observe the mortality of mud crabs. Thus, we selected 2.5mg/L cadmium. We have revise all section 3.6.
9.A list of abbreviation should be presented.
Response: Abbreviation is provided in the supplementary materials.
10.The legend of figure 2 should be revised.
Response: We have revised figure 2.
11.Figure 6: please label x-axis in the chart (h).
Response: We have revised figure 6.

Reviewer 2 Report
The Paper “Cadmium exposure induced hepatotoxicity in the mud crab (Scylla paramamosain): Activation of oxidative stress and Nrf2 signaling pathway” reports important data about the toxic effects of one of the most dangerous heavy metals in ecosystem.
The contents are informative, since the mechanisms of contaminants induced toxicity continue to be of interest due to the distribution of these elements in the environment and a variety of organisms.
In addition, the model system “Mud crab, Scylla paramamosain” represents an important commercial crustacean that accumulates several pollutants due to its habitat and diet. Nowadays it is important to understand the defense strategy triggered by against environmental stress.
The manuscript is well written, the problematic is well defined and the scientific approach is correct. Apart from a few small imperfections to correct.
Introduction, Authors should provide the data of Cd concentration in some polluted waters.
Material and methods “2.2. Cadmium exposure”, the authors reports the concentrations of the exposure related to LC50.
- The chosen concentrations should be compared to the cadmium concentrations found in polluted sites to understand how many times the order of magnitude is.
- Were water concentrations measured in lab conditions? If yes, please provide analytical method used to confirm water concentrations of Cd used for exposures in this study.
- Information about the concentration of the stock solution of the cadmium chloride used, the pH value of the solutions or if it has any color would be relevant. At the method there is no mention to this stock solution concentration?
Figure 2 capture. Enlargement and a bar should be reported.
There are some grammatical and tense mistakes throughout the manuscript. Authors should thoroughly revise them.
A revision of the chemical formulas is strongly recommended. For example, look at H2O2.
Apart from the above suggestions, I do not find any objection to giving my favorable opinion for the publication of this work in Antioxidant journal.
Author Response
We would like to express our sincere thanks to you for the constructive and positive comments. Point by point responses to the comments are listed below.
1.Introduction, Authors should provide the data of Cd concentration in some polluted waters.
Response: Thank you for your suggestions. We have added Cd concentration in some polluted waters (line 37-38).
2.Material and methods “2.2. Cadmium exposure”, the authors reports the concentrations of the exposure related to LC50. The chosen concentrations should be compared to the cadmium concentrations found in polluted sites to understand how many times the order of magnitude is.
Response: Thank you for your suggestions. According to previous studies, Cd levels of water in many areas of China can reach as high as 100-900 μg/L. In this study, we also set the Cd concentration related to LC50 and Cd concentration in some areas of China. Thus, we chosen 0.01 ((low), 0.05 ((middle) and 0.125 (high) mg/L Cd. We have added these in formation in the paper (line 37-38).
3.Were water concentrations measured in lab conditions? If yes, please provide analytical method used to confirm water concentrations of Cd used for exposures in this study.
Response: Thank you for your suggestions. We have measured the water concentrations of Cd. Water samples were acidified by HNO3. Cd concentrations were quantified by ICP-OES. We added this information in the paper(line 88-90).
4.Information about the concentration of the stock solution of the cadmium chloride used, the pH value of the solutions or if it has any color would be relevant. At the method there is no mention to this stock solution concentration?
Response: A stock solution of cadmium chloride (1g/L) was used as a source of the total cadmium level, which was subsequently diluted to the desired concentrations of cadmium. The stock solution of cadmium chloride is transparent. We added this information in the paper (line 83-85)
5.Figure 2 capture. Enlargement and a bar should be reported.
Response: Thank you for your suggestions. We have revised it.
6.There are some grammatical and tense mistakes throughout the manuscript. Authors should thoroughly revise them. A revision of the chemical formulas is strongly recommended. For example, look at H2O2. Apart from the above suggestions, I do not find any objection to giving my favorable opinion for the publication of this work in Antioxidant journal.
Response: Thank you for your suggestions. We have revised them in the paper.

Reviewer 3 Report
The subject of the manuscript is consistent with the scope of the Journal.
The abstract faithfully conveys the scope of investigations and conclusions drawn. The keywords correspond well to the scope of the research.
The aim of this study was to examine the effects of cadmium exposure on the antioxidant system and histopathological alterations were investigated. In order to understanding Nrf2 signaling pathway in cadmium-induced hepatotoxicity in the mud crab, Nrf2, CYP2 and MT expressions after cadmium exposure were examined. Furthermore, authors used RNAi technology to determine the function of Nrf2 signaling pathway in responses to cadmium exposure.
I think the paper is rather good. It needs some corrections:
1) add references to all analytical methods,
2) add some new references to Introduction and Discussion sections (the discussion is not very good now),
3) standardize References section.
Other sections are rather good.
Paper needs some editorial corrections (see: Instructions for Authors).
You must check your paper very exactly and correct all mistakes and complete lacking data of papers.
Author Response
We would like to express our sincere thanks to you for the constructive and positive comments. Point by point responses to the comments are listed below.
1.add references to all analytical methods,
Response: Thank you for your suggestions. We have added these references about analytical methods in the Materials and Methods section.。
2.add some new references to Introduction and Discussion sections (the discussion is not very good now),
Response: Thank you for your suggestions. We have revise them in the paper.。
3.standardize References section.
Response: We have revise them .
4.Paper needs some editorial corrections (see: Instructions for Authors).
You must check your paper very exactly and correct all mistakes and complete lacking data of papers.
Response: Thank you for your suggestions. We have revise them in the paper.

Round 2
Reviewer 1 Report
The authors answered the reviewer’s comments, and the paper has been improvement. I have no further comments.
Author Response
Thank you for your suggestions. We have revised them in the paper according the reviewer’s comments. We did not receive the second reviewer’s comments.